# Generalization of ERM in Stochastic Convex Optimization:
# The Dimension Strikes Back[*]

**Vitaly Feldman**
IBM Research – Almaden

## Abstract

In stochastic convex optimization the goal is to minimize a convex function
$F(x) \doteq \mathbf{E}_{\mathbf{f} \sim D}[\mathbf{f}(x)]$ over a convex set $\mathcal{K} \subset \mathbb{R}^d$ where $D$ is some unknown
distribution and each $f(\cdot)$ in the support of $D$ is convex over $\mathcal{K}$. The optimi-
zation is commonly based on i.i.d. samples $f^1, f^2, \ldots, f^n$ from $D$. A standard
approach to such problems is empirical risk minimization (ERM) that optimizes
$F_S(x) \doteq \frac{1}{n} \sum_{i \le n} f^i(x)$. Here we consider the question of how many samples
are necessary for ERM to succeed and the closely related question of uniform
convergence of $F_S$ to $F$ over $\mathcal{K}$. We demonstrate that in the standard $\ell_p/\ell_q$ setting
of Lipschitz-bounded functions over a $\mathcal{K}$ of bounded radius, ERM requires sample
size that scales linearly with the dimension $d$. This nearly matches standard upper
bounds and improves on $\Omega(\log d)$ dependence proved for $\ell_2/\ell_2$ setting in [18]. In
stark contrast, these problems can be solved using dimension-independent number
of samples for $\ell_2/\ell_2$ setting and $\log d$ dependence for $\ell_1/\ell_\infty$ setting using other
approaches.

We further show that our lower bound applies even if the functions in the support
of $D$ are smooth and efficiently computable and even if an $\ell_1$ regularization term is
added. Finally, we demonstrate that for a more general class of bounded-range (but
not Lipschitz-bounded) stochastic convex programs an infinite gap appears already
in dimension 2.

## 1 Introduction

Numerous central problems in machine learning, statistics and operations research are special cases of
stochastic optimization from i.i.d. data samples. In this problem the goal is to optimize the value of the
expected objective function $F(x) \doteq \mathbf{E}_{\mathbf{f} \sim D}[\mathbf{f}(x)]$ over some set $\mathcal{K}$ given i.i.d. samples $f^1, f^2, \ldots, f^n$
of $\mathbf{f}$. For example, in supervised learning the set $\mathcal{K}$ consists of hypothesis functions from $Z$ to $Y$
and each sample is an example described by a pair $(z, y) \in (Z, Y)$. For some fixed loss function
$L : Y \times Y \to \mathbb{R}$, an example $(z, y)$ defines a function from $\mathcal{K}$ to $\mathbb{R}$ given by $f_{(z,y)}(h) = L(h(z), y)$.
The goal is to find a hypothesis $h$ that (approximately) minimizes the expected loss relative to some
distribution $P$ over examples: $\mathbf{E}_{(z,y) \sim P}[L(h(z), y)] = \mathbf{E}_{(z,y) \sim P}[f_{(z,y)}(h)]$.

Here we are interested in stochastic convex optimization (SCO) problems in which $\mathcal{K}$ is some convex
subset of $\mathbb{R}^d$ and each function in the support of $D$ is convex over $\mathcal{K}$. The importance of this
setting stems from the fact that such problems can be solved efficiently via a large variety of known
techniques. Therefore in many applications even if the original optimization problem is not convex, it
is replaced by a convex relaxation.

A classic and widely-used approach to solving stochastic optimization problems is empirical risk
minimization (ERM) also referred to as stochastic average approximation (SAA) in the optimization

---

[*] See [9] for the full version of this work.

literature. In this approach, given a set of samples $S = (f^1, f^2, \ldots, f^n)$ the empirical objective function: $F_S(x) \doteq \frac{1}{n} \sum_{i \leq n} f^i(x)$ is optimized (sometimes with an additional regularization term such as $\lambda \|x\|^2$ for some $\lambda > 0$). The question we address here is the number of samples required for this approach to work *distribution-independently*. More specifically, for some fixed convex body $\mathcal{K}$ and fixed set of convex functions $\mathcal{F}$ over $\mathcal{K}$, what is the smallest number of samples $n$ such that for every probability distribution $D$ supported on $\mathcal{F}$, any algorithm that minimizes $F_S$ given $n$ i.i.d. samples from $D$ will produce an $\epsilon$-optimal solution $\hat{x}$ to the problem (namely, $F(\hat{x}) \leq \min_{x \in \mathcal{K}} F(x) + \epsilon$) with probability at least $1 - \delta$? We will refer to this number as the sample complexity of ERM for $\epsilon$-optimizing $\mathcal{F}$ over $\mathcal{K}$ (we will fix $\delta = 1/2$ for now).

The sample complexity of ERM for $\epsilon$-optimizing $\mathcal{F}$ over $\mathcal{K}$ is lower bounded by the sample complexity of $\epsilon$-optimizing $\mathcal{F}$ over $\mathcal{K}$, that is the number of samples that is necessary to find an $\epsilon$-optimal solution for any algorithm. On the other hand, it is upper bounded by the number of samples that ensures uniform convergence of $F_S$ to $F$. Namely, if with probability $\geq 1 - \delta$, for all $x \in \mathcal{K}$, $|F_S(x) - F(x)| \leq \epsilon/2$ then, clearly, any algorithm based on ERM will succeed. As a result, ERM and uniform convergence are the primary tool for analysis of the sample complexity of learning problems and are the key subject of study in statistical learning theory. Fundamental results in VC theory imply that in some settings, such as binary classification and least-squares regression, uniform convergence is also a necessary condition for learnability (*e.g.* [23, 17]) and therefore the three measures of sample complexity mentioned above nearly coincide.

In the context of stochastic convex optimization the study of sample complexity of ERM and uniform convergence was initiated in a groundbreaking work of Shalev-Shwartz, Shamir, Srebro and Sridharan [18]. They demonstrated that the relationships between these notions of sample complexity are substantially more delicate even in the most well-studied settings of SCO. Specifically, let $\mathcal{K}$ be a unit $\ell_2$ ball and $\mathcal{F}$ be the set of all convex sub-differentiable functions with Lipschitz constant relative to $\ell_2$ bounded by 1 or, equivalently, $\|\nabla f(x)\|_2 \leq 1$ for all $x \in \mathcal{K}$. Then, known algorithm for SCO imply that sample complexity of this problem is $O(1/\epsilon^2)$ and often expressed as $1/\sqrt{n}$ rate of convergence (*e.g.* [14, 17]). On the other hand, Shalev-Shwartz *et al.*[18] show[2] that the sample complexity of ERM for solving this problem with $\epsilon = 1/2$ is $\Omega(\log d)$. The only known upper bound for sample complexity of ERM is $\tilde{O}(d/\epsilon^2)$ and relies only on the uniform convergence of Lipschitz-bounded functions [21, 18].

As can seen from this discussion, the work of Shalev-Shwartz *et al.*[18] still leaves a major gap between known bounds on sample complexity of ERM (and also uniform convergence) for this basic Lipschitz-bounded $\ell_2/\ell_2$ setup. Another natural question is whether the gap is present in the popular $\ell_1/\ell_\infty$ setup. In this setup $\mathcal{K}$ is a unit $\ell_1$ ball (or in some cases a simplex) and $\|\nabla f(x)\|_\infty \leq 1$ for all $x \in \mathcal{K}$. The sample complexity of SCO in this setup is $\theta(\log d/\epsilon^2)$ (*e.g.* [14, 17]) and therefore, even an appropriately modified lower bound in [18], does not imply any gap. More generally, the choice of norm can have a major impact on the relationship between these sample complexities and hence needs to be treated carefully. For example, for (the reversed) $\ell_\infty/\ell_1$ setting the sample complexity of the problem is $\theta(d/\epsilon^2)$ (*e.g.* [10]) and nearly coincides with the number of samples sufficient for uniform convergence.

## 1.1 Overview of Results

In this work we substantially strengthen the lower bound in [18] proving that a linear dependence on the dimension $d$ is necessary for ERM (and, consequently, uniform convergence). We then extend the lower bound to all $\ell_p/\ell_q$ setups and examine several related questions. Finally, we examine a more general setting of bounded-range SCO (that is $|f(x)| \leq 1$ for all $x \in \mathcal{K}$). While the sample complexity of this setting is still low (for example $\tilde{O}(1/\epsilon^2)$ when $\mathcal{K}$ is an $\ell_2$ ball) and efficient algorithms are known, we show that ERM might require an infinite number of samples already for $d = 2$.

Our work implies that in SCO, even optimization algorithms that exactly minimize the empirical objective function can produce solutions with generalization error that is much larger than the generalization error of solutions obtained via some standard approaches. Another, somewhat counterintuitive, conclusion from our lower bounds is that, from the point of view of generalization of ERM and uniform convergence, convexity does not reduce the sample complexity in the worst case.

**Basic construction:** Our basic construction is fairly simple and its analysis is inspired by the technique in [18]. It is based on functions of the form $\max\{1/2, \max_{v \in V} \langle v, x \rangle\}$. Note that the maximum operator preserves both convexity and Lipschitz bound (relative to any norm). See Figure 1 for an illustration of such function for $d = 2$.

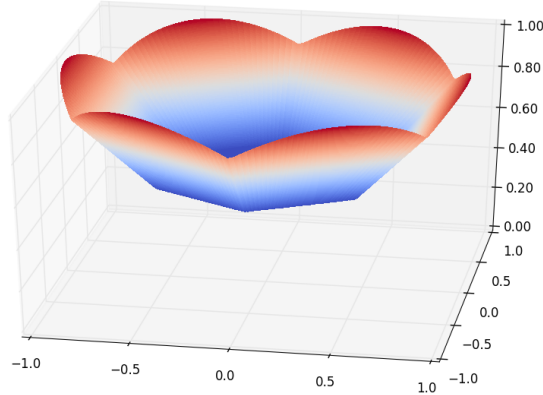

Figure 1: Basic construction for $d = 2$.

The distribution over the sets $V$ that define such functions is uniform over all subsets of some set of vectors $W$ of size $2^{d/6}$ such that for any two district $u, v \in W$, $\langle u, v \rangle \leq 1/2$. Equivalently, each element of $W$ is included in $V$ with probability $1/2$ independently of other elements in $W$. This implies that if the number of samples is less than $d/6$ then, with probability $> 1/2$, at least one of the vectors in $W$ (say $w$) will not be observed in any of the samples. This implies that $F_S$ can be minimized while maximizing $\langle w, x \rangle$ (the maximum over the unit $\ell_2$ ball is $w$). Note that a function randomly chosen from our distribution includes the term $\langle w, x \rangle$ in the maximum operator with probability $1/2$. Therefore the value of the expected function $F$ at $w$ is $3/4$ whereas the minimum of $F$ is $1/2$. In particular, there exists an ERM algorithm with generalization error of at least $1/4$. The details of the construction appear in Sec. 3.1 and Thm. 3.3 gives the formal statement of the lower bound. We also show that, by scaling the construction appropriately, we can obtain the same lower bound for any $\ell_p/\ell_q$ setup with $1/p + 1/q = 1$ (see Thm. 3.4).

**Low complexity construction:** The basic construction relies on functions that require $2^{d/6}$ bits to describe and exponential time to compute. Most application of SCO use efficiently computable functions and therefore it is natural to ask whether the lower bound still holds for such functions. To answer this question we describe a construction based on a set of functions where each function requires just $\log d$ bits to describe (there are at most $d/2$ functions in the support of the distribution) and each function can be computed in $O(d)$ time. To achieve this we will use $W$ that consists of (scaled) codewords of an asymptotically good and efficiently computable binary error-correcting code [12, 22]. The functions are defined in a similar way but the additional structure of the code allows to use at most $d/2$ subsets of $W$ to define the functions. Further details of the construction appear in Section 4.

**Smoothness:** The use of maximum operator results in functions that are highly non-smooth (that is, their gradient is not Lipschitz-bounded) whereas the construction in [18] uses smooth functions. Smoothness plays a crucial role in many algorithms for convex optimization (see [5] for examples). It reduces the sample complexity of SCO in $\ell_2/\ell_2$ setup to $O(1/\epsilon)$ when the smoothness parameter is a constant (*e.g.* [14, 17]). Therefore it is natural to ask whether our strong lower bound holds for smooth functions as well. We describe a modification of our construction that proves a similar lower bound in the smooth case (with generalization error of $1/128$). The main idea is to replace each linear function $\langle v, x \rangle$ with some smooth function $\nu(\langle v, x \rangle)$ guaranteing that for different vectors $v^1, v^2 \in W$ and every $x \in \mathcal{K}$, only one of $\nu(\langle v^1, x \rangle)$ and $\nu(\langle v^2, x \rangle)$ can be non-zero. This allows to easily control the smoothness of $\max_{v \in V} \nu(\langle v, x \rangle)$. See Figure 2 for an illustration of a function on which the construction is based (for $d = 2$). The details of this construction appear in Sec. 3.2 and the formal statement in Thm. 3.6.

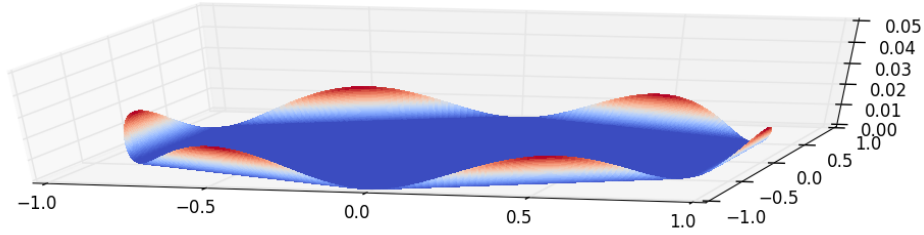

Figure 2: Construction using 1-smooth functions for $d = 2$.

$\ell_1$**-regularization:** Another important contribution in [18] is the demonstration of the important role that strong convexity plays for generalization in SCO: Minimization of $F_S(x) + \lambda R(x)$ ensures that ERM will have low generalization error whenever $R(x)$ is strongly convex (for a sufficiently large $\lambda$). This result is based on the proof that ERM of a strongly convex Lipschitz function is *uniform replace-one* stable and the connection between such stability and generalization showed in [4] (see also [19] for a detailed treatment of the relationship between generalization and stability). It is natural to ask whether other approaches to regularization will ensure generalization. We demonstrate that for the commonly used $\ell_1$ regularization the answer is negative. We prove this using a simple modification of our lower bound construction: We shift the functions to the positive orthant where the regularization terms $\lambda\|x\|_1$ is just a linear function. We then subtract this linear function from each function in our construction, thereby balancing the regularization (while maintaining convexity and Lipschitz-boundedness). The details of this construction appear in Sec. 3.3 (see Thm. 3.7).

**Dependence on accuracy:** For simplicity and convenience we have ignored the dependence on the accuracy $\epsilon$, Lipschitz bound $L$ and radius $R$ of $\mathcal{K}$ in our lower bounds. It is easy to see, that this more general setting can be reduced to the case we consider here (Lipschitz bound and radius are equal to 1) with accuracy parameter $\epsilon' = \epsilon/(LR)$. We generalize our lower bound to this setting and prove that $\Omega(d/\epsilon'^2)$ samples are necessary for uniform convergence and $\Omega(d/\epsilon')$ samples are necessary for generalization of ERM. Note that the upper bound on the sample complexity of these settings is $\tilde{O}(d/\epsilon'^2)$ and therefore the dependence on $\epsilon'$ in our lower bound does not match the upper bound for ERM. Resolving this gap or even proving any $\omega(d/\epsilon' + 1/\epsilon'^2)$ lower bound is an interesting open problem. Additional details can be found in the full version.

**Bounded-range SCO:** Finally, we consider a more general class of bounded-range convex functions Note that the Lipschitz bound of 1 and the bound of 1 on the radius of $\mathcal{K}$ imply a bound of 1 on the range (up to a constant shift which does not affect the optimization problem). While this setting is not as well-studied, efficient algorithms for it are known. For example, the online algorithm in a recent work of Rakhlin and Sridharan [16] together with standard online-to-batch conversion arguments [6], imply that the sample complexity of this problem is $\tilde{O}(1/\epsilon^2)$ for any $\mathcal{K}$ that is an $\ell_2$ ball (of any radius). For general convex bodies $\mathcal{K}$, the problems can be solved via random walk-based approaches [3, 10] or an adaptation of the center-of-gravity method given in [10]. Here we show that for this setting ERM might completely fail already for $\mathcal{K}$ being the unit 2-dimensional ball. The construction is based on ideas similar to those we used in the smooth case and is formally described in in the full version.

## 2   Preliminaries

For an integer $n \geq 1$ let $[n] \doteq \{1, \ldots, n\}$. Random variables are denoted by bold letters, e.g., $\mathbf{f}$. Given $p \in [1, \infty]$ we denote the ball of radius $R > 0$ in $\ell_p$ norm by $\mathcal{B}_p^d(R)$, and the unit ball by $\mathcal{B}_p^d$.

For a convex body (i.e., compact convex set with nonempty interior) $\mathcal{K} \subseteq \mathbb{R}^d$, we consider problems of the form

$$\min_{\mathcal{K}}(F_D) \doteq \min_{x \in \mathcal{K}} \left\{ F_D(x) \doteq \mathop{\mathbf{E}}_{\mathbf{f} \sim D}[\mathbf{f}(x)] \right\},$$

where $\mathbf{f}$ is a random variable defined over some set of convex, sub-differentiable functions $\mathcal{F}$ on $\mathcal{K}$ and distributed according to some unknown probability distribution $D$. We denote $F^* = \min_{\mathcal{K}}(F_D)$. For an approximation parameter $\epsilon > 0$ the goal is to find $x \in \mathcal{K}$ such that $F_D(x) \leq F^* + \epsilon$ and we call any such $x$ an $\epsilon$-*optimal solution*. For an $n$-tuple of functions $S = (f^1, \ldots, f^n)$ we denote by $F_S \doteq \frac{1}{n} \sum_{i \in [n]} f^i$.

We say that a point $\hat{x}$ is an empirical risk minimum for an $n$-tuple $S$ of functions over $\mathcal{K}$, if $F_S(\hat{x}) = \min_{\mathcal{K}}(F_S)$. In some cases there are many points that minimize $F_S$ and in this case we refer to a specific algorithm that selects one of the minimums of $F_S$ as an empirical risk minimizer. To make this explicit we refer to the output of such a minimizer by $\hat{x}(S)$ .

Given $x \in \mathcal{K}$, and a convex function $f$ we denote by $\nabla f(x) \in \partial f(x)$ an arbitrary selection of a subgradient. Let us make a brief reminder of some important classes of convex functions. Let $p \in [1, \infty]$ and $q = p_* \doteq 1/(1 - 1/p)$. We say that a subdifferentiable convex function $f : \mathcal{K} \to \mathbb{R}$ is in the class

- $\mathcal{F}(\mathcal{K}, B)$ of $B$-bounded-range functions if for all $x \in \mathcal{K}$, $|f(x)| \leq B$.
- $\mathcal{F}_p^0(\mathcal{K}, L)$ of $L$-Lipschitz continuous functions w.r.t. $\ell_p$, if for all $x, y \in \mathcal{K}$, $|f(x) - f(y)| \leq L\|x - y\|_p$;
- $\mathcal{F}_p^1(\mathcal{K}, \sigma)$ of functions with $\sigma$-Lipschitz continuous gradient w.r.t. $\ell_p$, if for all $x, y \in \mathcal{K}$, $\|\nabla f(x) - \nabla f(y)\|_q \leq \sigma\|x - y\|_p$.

We will omit $p$ from the notation when $p = 2$. Omitted proofs can be found in the full version [9].

## 3 Lower Bounds for Lipschitz-Bounded SCO

In this section we present our main lower bounds for SCO of Lipschitz-bounded convex functions. For comparison purposes we start by formally stating some known bounds on sample complexity of solving such problems. The following uniform convergence bounds can be easily derived from the standard covering number argument (*e.g.* [21, 18])

**Theorem 3.1.** *For $p \in [1, \infty]$, let $\mathcal{K} \subseteq \mathcal{B}_p^d(R)$ and let $D$ be any distribution supported on functions $L$-Lipschitz on $\mathcal{K}$ relative to $\ell_p$ (not necessarily convex). Then, for every $\epsilon, \delta > 0$ and $n \geq n_1 = O\left(\frac{d \cdot (LR)^2 \cdot \log(dLR/(\epsilon\delta))}{\epsilon^2}\right)$*

$$\Pr_{\mathbf{S} \sim D^n} [\exists x \in \mathcal{K}, \ |F_D(x) - F_{\mathbf{S}}(x)| \geq \epsilon] \leq \delta.$$

The following upper bounds on sample complexity of Lipschitz-bounded SCO can be obtained from several known algorithms [14, 18] (see [17] for a textbook exposition for $p = 2$).

**Theorem 3.2.** *For $p \in [1, 2]$, let $\mathcal{K} \subseteq \mathcal{B}_p^d(R)$. Then, there is an algorithm $\mathcal{A}_p$ that given $\epsilon, \delta > 0$ and $n = n_p(d, R, L, \epsilon, \delta)$ i.i.d. samples from any distribution $D$ supported on $\mathcal{F}_p^0(\mathcal{K}, L)$, outputs an $\epsilon$-optimal solution to $F_D$ over $\mathcal{K}$ with probability $\geq 1 - \delta$. For $p \in (1, 2]$, $n_p = O((LR/\epsilon)^2 \cdot \log(1/\delta))$ and for $p = 1$, $n_p = O((LR/\epsilon)^2 \cdot \log d \cdot \log(1/\delta))$.*

Stronger results are known under additional assumptions on smoothness and/or strong convexity (*e.g.* [14, 15, 20, 1]).

### 3.1 Non-smooth construction

We will start with a simpler lower bound for non-smooth functions. For simplicity, we will also restrict $R = L = 1$. Lower bounds for the general setting can be easily obtained from this case by scaling the domain and desired accuracy.

We will need a set of vectors $W \subseteq \{-1, 1\}^d$ with the following property: for any distinct $w^1, w^2 \in W$, $\langle w^1, w^2 \rangle \leq d/2$. The Chernoff bound together with a standard packing argument imply that there exists a set $W$ with this property of size $\geq e^{d/8} \geq 2^{d/6}$.

For any subset $V$ of $W$ we define a function

$$g_V(x) \doteq \max\{1/2, \max_{\bar{w} \in V}\langle \bar{w}, x \rangle\}, \tag{1}$$

where $\bar{w} \doteq w/\|w\| = w/\sqrt{d}$. See Figure 1 for an illustration. We first observe that $g_V$ is convex and 1-Lipschitz (relative to $\ell_2$). This immediately follows from $\langle \bar{w}, x \rangle$ being convex and 1-Lipschitz for every $w$ and $g_V$ being the maximum of convex and 1-Lipschitz functions.

**Theorem 3.3.** *Let $\mathcal{K} = \mathcal{B}_2^d$ and we define $\mathcal{H}_2 \doteq \{g_V \mid V \subseteq W\}$ for $g_V$ defined in eq. (1). Let $D$ be the uniform distribution over $\mathcal{H}_2$. Then for $n \leq d/6$ and every set of samples $S$ there exists an ERM $\hat{x}(S)$ such that*

$$\Pr_{\mathbf{S} \sim D^n} [F_D(\hat{x}(\mathbf{S})) - F^* \geq 1/4] > 1/2.$$

*Proof.* We start by observing that the uniform distribution over $\mathcal{H}_2$ is equivalent to picking the function $g_{\mathbf{V}}$ where $\mathbf{V}$ is obtained by including every element of $W$ with probability $1/2$ randomly and independently of all other elements. Further, by the properties of $W$, for every $w \in W$, and $V \subseteq W$, $g_V(\bar{w}) = 1$ if $w \in V$ and $g_V(\bar{w}) = 1/2$ otherwise. For $g_{\mathbf{V}}$ chosen randomly with respect to $D$, we have that $w \in \mathbf{V}$ with probability exactly $1/2$. This implies that $F_D(\bar{w}) = 3/4$.

Let $\mathbf{S} = (g_{\mathbf{V}_1}, \ldots, g_{\mathbf{V}_n})$ be the random samples. Observe that $\min_{\mathcal{K}}(F_{\mathbf{S}}) = 1/2$ and $F^* = \min_{\mathcal{K}}(F_D) = 1/2$ (the minimum is achieved at the origin $\bar{0}$). Now, if $\bigcup_{i \in [n]} \mathbf{V}_i \neq W$ then let $\hat{x}(\mathbf{S}) \doteq \bar{w}$ for any $w \in W \setminus \bigcup_{i \in [n]} \mathbf{V}_i$. Otherwise $\hat{x}(\mathbf{S})$ is defined to be the origin $\bar{0}$. Then by the property of $\mathcal{H}_2$ mentioned above, we have that for all $i$, $g_{\mathbf{V}_i}(\hat{x}(\mathbf{S})) = 1/2$ and hence $F_{\mathbf{S}}(\hat{x}(\mathbf{S})) = 1/2$. This means that $\hat{x}(\mathbf{S})$ is a minimizer of $F_{\mathbf{S}}$.

Combining these statements, we get that, if $\bigcup_{i \in [n]} \mathbf{V}_i \neq W$ then there exists an ERM $\hat{x}(\mathbf{S})$ such that $F_{\mathbf{S}}(\hat{x}(\mathbf{S})) = \min_{\mathcal{K}}(F_{\mathbf{S}})$ and $F_D(\hat{x}(\mathbf{S})) - F^* = 1/4$. Therefore to prove the claim it suffices to show that for $n \leq d/6$ we have that

$$\Pr_{\mathbf{S} \sim D^n} \left[ \bigcup_{i \in [n]} \mathbf{V}_i \neq W \right] > \frac{1}{2}.$$

This easily follows from observing that for the uniform distribution over subsets of $W$, for every $w \in W$,

$$\Pr_{\mathbf{S} \sim D^n} \left[ w \in \bigcup_{i \in [n]} \mathbf{V}_i \right] = 1 - 2^{-n}$$

and this event is independent from the inclusion of other elements in $\bigcup_{i \in [n]} \mathbf{V}_i$. Therefore

$$\Pr_{\mathbf{S} \sim D^n} \left[ \bigcup_{i \in [n]} \mathbf{V}_i = W \right] = \left( 1 - 2^{-n} \right)^{|W|} \leq e^{-2^{-n} \cdot 2^{d/6}} \leq e^{-1} < \frac{1}{2}.$$

$\square$

**Other $\ell_p$ norms:** We now observe that exactly the same approach can be used to extend this lower bound to $\ell_p/\ell_q$ setting. Specifically, for $p \in [1, \infty]$ and $q = p_*$ we define

$$g_{p,V}(x) \doteq \max \left\{ \frac{1}{2}, \max_{w \in V} \frac{\langle w, x \rangle}{d^{1/q}} \right\}.$$

It is easy to see that for every $V \subseteq W$, $g_{q,V} \in \mathcal{F}_p^0(\mathcal{B}_p^d, 1)$. We can now use the same argument as before with the appropriate normalization factor for points in $\mathcal{B}_p^d$. Namely, instead of $\bar{w}$ for $w \in W$ we consider the values of the minimized functions at $w/d^{1/p} \in \mathcal{B}_p^d$. This gives the following generalization of Thm. 3.3.

**Theorem 3.4.** *For every $p \in [1, \infty]$ let $\mathcal{K} = \mathcal{B}_p^d$ and we define $\mathcal{H}_p \doteq \{g_{p,V} \mid V \subseteq W\}$ and let $D$ be the uniform distribution over $\mathcal{H}_p$. Then for $n \leq d/6$ and every set of samples $S$ there exists an ERM $\hat{x}(S)$ such that*

$$\Pr_{\mathbf{S} \sim D^n} [F_D(\hat{x}(\mathbf{S})) - F^* \geq 1/4] > 1/2.$$

## 3.2 Smoothness does not help

We now extend the lower bound to smooth functions. We will for simplicity restrict our attention to $\ell_2$ but analogous modifications can be made for other $\ell_p$ norms. The functions $g_V$ that we used in the construction use two maximum operators each of which introduces non-smoothness. To deal with maximum with $1/2$ we simply replace the function $\max\{1/2, \langle \bar{w}, x \rangle\}$ with a quadratically smoothed version (in the same way as hinge loss is sometimes replaced with modified Huber loss). To deal with the maximum over all $w \in V$, we show that it is possible to ensure that individual components do not "interact". That is, at every point $x$, the value, gradient and Hessian of at most one component function are non-zero (value, vector and matrix, respectively). This ensures that maximum becomes addition and Lipschitz/smoothness constants can be upper-bounded easily.

Formally, we define

$$\nu(a) \doteq \left\{ \begin{array}{ll} 0 & \text{if } a \leq 0 \\ a^2 & \text{otherwise.} \end{array} \right.$$

Now, for $V \subseteq W$, we define

$$h_V(x) \doteq \sum_{w \in V} \nu(\langle \bar{w}, x \rangle - 7/8). \tag{2}$$

See Figure 2 for an illustration. We first prove that $h_V$ is $1/4$-Lipschitz and $1$-smooth.

**Lemma 3.5.** *For every $V \subseteq W$ and $h_V$ defined in eq. (2) we have $h_V \in \mathcal{F}_2^0(\mathcal{B}_2^d, 1/4) \cap \mathcal{F}_2^1(\mathcal{B}_2^d, 1)$.*

From here we can use the proof approach from Thm. 3.3 but with $h_V$ in place of $g_V$.

**Theorem 3.6.** *Let $\mathcal{K} = \mathcal{B}_2^d$ and we define $\mathcal{H} \doteq \{h_V \mid V \subseteq W\}$ for $h_V$ defined in eq. (2). Let $D$ be the uniform distribution over $\mathcal{H}$. Then for $n \leq d/6$ and every set of samples $S$ there exists an ERM $\hat{x}(S)$ such that*

$$\Pr_{\mathbf{S} \sim D^n} \left[ F_D(\hat{x}(\mathbf{S})) - F^* \geq 1/128 \right] > 1/2.$$

## 3.3 $\ell_1$ Regularization does not help

Next we show that the lower bound holds even with an additional $\ell_1$ regularization term $\lambda \|x\|$ for positive $\lambda \leq 1/\sqrt{d}$. (Note that if $\lambda > 1/\sqrt{d}$ then the resulting program is no longer 1-Lipschitz relative to $\ell_2$. Any constant $\lambda$ can be allowed for $\ell_1/\ell_\infty$ setup). To achieve this we shift the construction to the positive orthant (that is $x$ such that $x_i \geq 0$ for all $i \in [d]$). In this orthant the subgradient of the regularization term is simply $\lambda \bar{1}$ where $\bar{1}$ is the all 1's vector. We can add a linear term to each function in our distribution that balances this term thereby reducing the analysis to non-regularized case. More formally, we define the following family of functions. For $V \subseteq W$,

$$h_V^\lambda(x) \doteq h_V(x - \bar{1}/\sqrt{d}) - \lambda \langle \bar{1}, x \rangle. \tag{3}$$

Note that over $\mathcal{B}_2^d(2)$, $h_V^\lambda(x)$ is $L$-Lipschitz for $L \leq 2(2 - 7/8) + \lambda\sqrt{d} \leq 9/4$. We now state and prove this formally.

**Theorem 3.7.** *Let $\mathcal{K} = \mathcal{B}_2^d(2)$ and for a given $\lambda \in (0, 1/\sqrt{d}]$, we define $\mathcal{H}^\lambda \doteq \{h_V^\lambda \mid V \subseteq W\}$ for $h_V^\lambda$ defined in eq. (3). Let $D$ be the uniform distribution over $\mathcal{H}^\lambda$. Then for $n \leq d/6$ and every set of samples $S$ there exists $\hat{x}(S)$ such that*

- *$F_S(\hat{x}(S)) = \min_{x \in \mathcal{K}}(F_S(x) + \lambda \|x\|_1)$;*

- *$\Pr_{\mathbf{S} \sim D^n} \left[ F_D(\hat{x}(\mathbf{S})) - F^* \geq 1/128 \right] > 1/2$.*

## 4 Lower Bound for Low-Complexity Functions

We will now demonstrate that our lower bounds hold even if one restricts the attention to functions that can be computed efficiently (in time polynomial in $d$). For this purpose we will rely on known constructions of binary linear error-correcting codes. We describe the construction for non-smooth $\ell_2/\ell_2$ setting but analogous versions of other constructions can be obtained in the same way.

We start by briefly providing the necessary background about binary codes. For two vectors $w^1, w^2 \in \{\pm 1\}^d$ let $\#_{\neq}(w^1, w^2)$ denote the Hamming distance between the two vectors. We say that a mapping $G : \{\pm 1\}^k \to \{\pm 1\}^d$ is a $[d, k, r, T]$ binary error-correcting code if $G$ has distance at least $2r + 1$, $G$ can be computed in time $T$ and there exists an algorithm that for every $w \in \{\pm 1\}^d$ such that for some $z \in \{\pm 1\}^k$, $\#_{\neq}(w, G(z)) \leq r$ finds such $z$ in time $T$ (note that such $z$ is unique).

Given $[d, k, r, T]$ code $G$, for every $j \in [k]$, we define a function

$$g_j(x) \doteq \max \left\{ 1 - \frac{r}{2d}, \max_{w \in W_j} \langle \bar{w}, x \rangle \right\}, \tag{4}$$

where $W_j \doteq \{G(z) \mid z \in \{\pm 1\}^k, z_j = 1\}$. As before, we note that $g_j$ is convex and 1-Lipschitz (relative to $\ell_2$).

We can now use any existing constructions of efficient binary error-correcting codes to obtain a lower bound that uses only a small set of efficiently computable convex functions. Getting a lower bound that has asymptotically optimal dependence on $d$ requires that $k = \Omega(d)$ and $r = \Omega(d)$ (referred to as being *asymptotically good*). The existence of efficiently computable and asymptotically good binary error-correcting codes was first shown by Justesen [12]. More recent work of Spielman [22] shows existence of asymptotically good codes that can be encoded and decoded in $O(d)$ time. In particular, for some constant $\rho > 0$, there exists a $[d, d/2, \rho \cdot d, O(d)]$ binary error-correcting code. As a corollary we obtain the following lower bound.

**Corollary 4.1.** *Let $G$ be an asymptotically-good $[d, d/2, \rho \cdot d, O(d)]$ error-correcting code for a constant $\rho > 0$. Let $\mathcal{K} = \mathcal{B}_2^d$ and we define $\mathcal{H}_G \doteq \{g_j \mid j \in [d/2]\}$ for $g_j$ defined in eq. (4). Let $D$ be the uniform distribution over $\mathcal{H}_G$. Then for every $x \in \mathcal{K}$, $g_j(x)$ can be computed in time $O(d)$. Further, for $n \leq d/4$ and every set of samples $S \in \mathcal{H}_G^n$ there exists an ERM $\hat{x}(S)$ such that*

$$F_D(\hat{x}(S)) - F^* \geq \rho/4.$$

## 5   Discussion

Our work points out to substantial limitations of the classic approach to understanding and analysis of generalization in the context of general SCO. Further, it implies that in order to understand how well solutions produced by an optimization algorithm generalize, it is necessary to examine the optimization algorithm itself. This is a challenging task that we still have relatively few tools to address. Yet such understanding is also crucial for developing theory to guide the design of optimization algorithms that are used in machine learning applications.

One way to bypass our lower bounds is to use additional structural assumptions. For example, for generalized linear regression problems uniform convergence gives nearly optimal bounds on sample complexity [13]. One natural question is whether there exist more general classes of functions that capture most of the practically relevant SCO problems and enjoy dimension-independent (or, scaling as $\log d$) uniform convergence bounds.

An alternative approach is to bypass uniform convergence (and possibly also ERM) altogether. Among a large number of techniques that have been developed for ensuring generalization, the most general ones are based on notions of stability [4, 19]. However, known analyses based on stability often do not provide the strongest known generalization guarantees (*e.g.* high probability bounds require very strong assumptions). Another issue is that we lack general algorithmic tools for ensuring stability of the output. Therefore many open problems remain and significant progress is required to obtain a more comprehensive understanding of this approach. Some encouraging new developments in this area are the use of notions of stability derived from differential privacy [7, 8, 2] and the use of techniques for analysis of convergence of convex optimization algorithms for proving stability [11].

## Acknowledgements

I am grateful to Ken Clarkson, Sasha Rakhlin and Thomas Steinke for discussions and insightful comments related to this work.

## Footnotes

[2]The dependence on $d$ is not stated explicitly but follows immediately from their analysis.

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
