[Reviews · NeurIPS 2016]

Reviewer 1

Summary

The paper considers the problem of stochastic convex optimization over convex subset K of R^d. In this problem, the learner is getting convex functions f_1,...,f_m:K->R sampled from some distribution D. Its goal is to come up with a point x in K such that F(x):=E_f[f(x)] is as small as possible (the expectation is w.r.t. the distribution from which the functions were sampled). Using online convex optimization algorithms, it is possible to find x that is eps-optimal (i.e., satisfies F(x)< min_zF(z)+eps) using m = (LB)/eps^2 examples (Here, L is a bound on the Liphshitzness of the functions and B is the diameter of K). This differs from more classical statistical complexity results, in the sense that it is algorithm dependent and does not hold for any ERM algorithm (i.e. one that finds x in K that empirically does best on the sample). The paper revolves around the question of what can be achieved by any ERM algorithm. Previous work has shown that in that case there is additional multiplicative log(d) factor in the sample complexity for this setting. This discovery was very surprising, since, as noted before, classical upper bounds apply to all ERMs. The current paper substantially strengthen this result, and shows that there is actually additional multiplicative d factor in the sample complexity for arbitrary ERM. The authors further extend their ides and show similar bounds even if one assumes that the functions are smooth. Furthermore, if one drops the Liphshitzness assumption and replaces it by boundness assumption, the sample complexity becomes infinity already in d=2 (whereas specific algorithms have finite sample complexity). Surprisingly maybe, all proofs are clean and simple.

Qualitative Assessment

I think that the paper has several substantial contributions. First, it essentially determines the sample complexity of the problem they consider, which is very basic. Second, in my opinion, this is the most vital example of gaps between optimal algorithms and ERMs. Whereas in previous such results gaps were logarithmic, in this work the gap is linear! Lastly, due to the simplicity of the technical content, I believe that this result will prove very useful for further investigations of this topic, and will become a standard result in the area.

Confidence in this Review

3-Expert (read the paper in detail, know the area, quite certain of my opinion)


Reviewer 2

Summary

This paper provides two lower bounds (theorems 3.3 and 3.4) on the sample complexity of bounded-range 1-Lipschitz functions for the empirical risk minimization (ERM) approach to stochastic convex optimization. These lower bounds scale linearly on the dimension d of the space of predictors, thus improving on the log(d) dependence of the lower bound of ref 14. They also show (theorem 3.7) that an L1 regularizer fails to remove the linear dependence on d of the lower bound. Finally, they study the more general class of bounded range fonctions and show the the ERM approach can fail as soon as d >= 2.

Qualitative Assessment

This paper is reasonably-well presented apart from the fact that the overview (section 1.1) is longer than necessary and hard to grasp a priori. I would suggest reducing this part and enlarging the discussion a posteriori. Also, the notation used is not the standard one in machine learning (ML) (the usage of x instead of w and the usage of f_i instead of training points (x_i,y_i)); which makes it harder to read for the ML community. The paper is technically very strong. However, I am very concerned about the level of relevance of this paper to machine learning. Normally, each function f_i is induced by a training example (x_i, y_i) and the space of predictors coincide with the instance space \Xcal: the space in which each x_i is sampled (note that x_i could be the output of a deep neural net). But it appears to me that all the different constructions used in this paper to represent the functions f_i, (namely eq 1, eq 2, eq 3, and eq 4) cannot by realized by a training point (ie, a single training example) and a standard loss function. If this is the case, this means that all the theorems provided in this paper are irrelevant to machine learning. This explains why I have rated this paper at the substandard-for-nips level for impact and usefulness (point 7). I would be happy to change my score if the authors could provide clear examples of how these constructions can be achieved with a single training point and a standard loss function. ----- After the rebuttal phase: following the author's rebuttal, I have increased my score for potential impact.

Confidence in this Review

1-Less confident (might not have understood significant parts)


Reviewer 3

Summary

The paper gives lower bounds on the sample complexity of ERM. In particular, it shows how convex functions from two classes of (non-smooth and smooth) 1 Lipschitz functions, suffer constant generalization error when estimated using ERM given only d/6 samples. An additional elaboration of the construction shows that l_1 regularization does not help (by moving the function class to where the regularizer reduces to a linear functional). The last variant relates to functions that are not 1 Lipschitz, but that have bounded range. In this context, constant generalization error occurs even in dimension 2. Each function has a different subset of m disjoint "bumps".

Qualitative Assessment

Clarity: l.86: does size 2^d/6 refer to W or to its subsets? ambiguous. Rest of paragraph is not quite clear. l.103: spelling "guaranteing" l.104 typo: v^1 appears twice, v^2 is missing. Also, quantifiers unclear, maybe you mean that for ever x, only one of the functions can be non-zero? Proof of Thm 4.2 is a bit terse for comfort. Novelty: I am not aware of such bounds on ERM, but this is not my area. Technical quality: The results are given well, from clear Theorems, to a relation of these lower bounds to existing lower and upper bounds. Proofs appear sound. Impact: Lower bounds are very important, but more so when the construction is not too extreme. As the paper also states, it would not be surprised if upper bounds for rich but more natural classes eventually exceed some of these lower bounds.

Confidence in this Review

1-Less confident (might not have understood significant parts)


Reviewer 4

Summary

The paper considers the the question of when empirical risk minimization suffices to to actually find a true risk minimizer, a question which is fundamental to many statistical tasks. Rigorously, the setup is as follows: we wish to minimize E[f] where f is drawn from some distribution D, assuming the f's are "nice" in some sense (here they consider various settings, including convex, Lipschitz, etc). We get samples from f_1, ..., f_n from this distribution, and can form the empirical risk, which is the average of the sample f's. The question is whether or not minimizing this empirical risk will always get us a minimizer for the ground truth, or something close. In this work the authors provide strong lower bounds for this problem, in a variety of settings. Their most basic result is that for convex, Lipschitz functions over the unit l2 ball, to ensure that the ERM is O(1) close to the true minimizer requires Omega(d) samples, where di is the dimension. This is a tight lower bound in terms of the dimension. To the best of my knowledge, the previous best lower bounds were surprisingly weak--only Omega (log d). They can also modify their construction to show that a similar lower bound also holds other lp norms, for smooth, convex, Lipschitz functions, and for l1 regularization. Finally, a further modification shows that ERM never works for range constrained stochastic convex optimization (without a Lipschitz constraint), although other techniques work for this setting.

Qualitative Assessment

I like this result quite a bit. In my opinion, the question they consider is quite important. The lower bounds seem to give the right tradeoffs for different lp norms (at least, in terms of dependence on d). The reductions are well presented and clear, and I don't see any issue with them. The main drawback of their approach I see is that it does not easily generalize to also obtain tight bounds in terms of eps, the desired error. Moreover, often times (as they point out) algorithms can bypass the ERM entirely and get better sample complexities, so perhaps ERM is simply not the right quantity to study. However, since it is so pervasive in practice, I feel like ultimately understanding generalization of ERM is an important question and the authors make an important contribution to it, and for that reason I think it is worthy of inclusion in NIPS.

Confidence in this Review

2-Confident (read it all; understood it all reasonably well)


Reviewer 5

Summary

This paper provides a nearly tight lower bound on the sample complexity (or, equivalently the true risk) of the ERM approach in the convex, Lipschitz-bounded learning model. This makes a big step towards fully answering an important open question that remained open for almost 7 years. The main result shows that the sample complexity of the ERM approach for some convex, Lipschitz-bounded problem in $\ell_p/\ell_q$ must be at least linear in the dimensionality of the parameter space (i.e., the hypothesis class). This lower bound nearly matches the upper bound of $\tilde{O}(d/\epsilon^2)$, where $\epsilon$ is the true risk, that was obtained (for the case of p=2) in the work of Shalev-Schwartz, Shamir, Srebro, Sridharan in 2009. The best known lower bound before the result in this submission was ~$\Omega(\log(d)/\epsilon^2)$ derived for the case of $\ell_2/\ell_2$ in the same 2009 paper mentioned above. Moreover, it is shown that smoothness does not actually help as a similar lower bound can be derived for convex smooth loss functions. The authors also demonstrate that for the more general class of convex bounded functions, ERM may not work. In particular, it is shown for some problem in this setting, ERM does not have finite sample complexity.

Qualitative Assessment

Excellent work and interesting results of significant potential. The paper takes a big step towards a more complete characterization of the limits of the ERM approach for stochastic convex optimization and statistical learning. I am curious if the construction in the lower bound can be modified (without too much work) to give the "right" dependence on $\epsilon$ (that is, $d/\epsilon^2$). It would be great if the authors elaborate on that.

Confidence in this Review

3-Expert (read the paper in detail, know the area, quite certain of my opinion)


Reviewer 6

Summary

This paper provides new lower bounds to stochastic convex optimization problems with ERM. The key insight in this paper is: there are problems which are not learnable through ERM with samples sublinear in dimension, whereas such problems can be solved using dimension-independent number of samples (or logarithmic in dimension) by other approaches.

Qualitative Assessment

I apologize that I'm not an expert of the lower bounds on learning problems, and as a result I cannot give too many constructive comments. This paper is clearly written. I've gone through the proofs up to section 4 and everything seems to be correct. The ideas are simple and well presented. Intro is a little too lengthy for me, but it's also fine. The overall quality of this paper is good. My only concern about this paper is the novelty. Since I did not read many papers in this area, it is difficult for me to judge the quality of the ideas here, as well as the significance of their implications. I'll have to leave this to the editor or the more knowledgable reviewers. Some minor comments: 1. In the abstract/intro the authors woite: "The optimization is based on iid samples..." This is not always true: There are applications of Stochastic Convex Optimization which are not based on iid samples, and yet we still care about the expected loss. Although sample average approximation is the most popular scheme in machine learning, I would suggest rewriting the sentence for generality. 2. A typo in the first sentence in line 83.

Confidence in this Review

2-Confident (read it all; understood it all reasonably well)